# How Strategic Planning Enhances ESG: Evidence from Mission Statements

**Arafat Aljebrini [1,\*]**, **Kagan Dogruyol [1]** and **Ibraheem Y. Y. Ahmaro [2]**

[1]  Department of Business Administration, Faculty of Economics and Administrative Sciences, Cyprus International University, Haspolat Campus, Northern Cyprus, Via Mersin 10, 99040 Haspolat, Turkey; kdogruyol@ciu.edu.tr

[2]  Computer Science Department, College of Information Technology, Hebron University, Hebron P.O. Box 40, Palestine; ahmaro@hebron.edu

[\*]  Correspondence: arafatj@hebron.edu

**Abstract:** The purpose of this study is to analyze the mission statements of 49 companies listed on the Palestine Exchange (PEX), focusing on their structure and alignment with environmental, social, and governance (ESG) principles. The primary objective of this research was to evaluate the extent to which Palestinian companies embed sustainability issues into their mission statements. This was done with a qualitative research design and a descriptive content analysis method, letting mission statements from different fields be looked at in excellent detail. This analysis offers valuable insights into how Palestinian companies articulate their strategic goals and communicate their commitment to ESG factors. The findings reveal that Palestinian companies demonstrate a clear understanding of sustainability and its relevance to their operations. A lot of companies are also working hard to include sustainability principles in their mission statements. This shows that people are becoming more aware of how important ESG factors are for shaping business strategy and creating long-term value.

**Keywords:** strategic planning; mission statement; ESG; sustainability; PEX

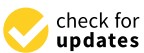

## 1. Introduction

Most of today's businesses strongly emphasize the importance of a mission statement to ensure long-term company sustainability. Despite their frequent usage, these concepts can sometimes be the most badly comprehended [1]. Peter Drucker's 1973 study emphasizes the lack of sufficient consideration given to the concept of mission in business, which he argues is a significant factor contributing to dissatisfaction and disappointment in the world of business. Drucker's statement remains as valid today as it was at the time [2–4]. As a result of the significant financial and economic limitations caused by the recent severe economic crisis, they are increasingly forced to create plans to enhance their efficiency and long-term viability in order to meet their objectives.

To begin, it is important to define the key concepts explored in this study. A company's mission represents its core purpose and reason for being. A mission statement should address essential questions, including the organization's objectives and its purpose. Such statements are widely regarded as vital for a firm's long-term success and strategic direction, as they play a crucial role in influencing performance and promoting sustainable growth.

However, mission statements often receive limited focus in organizational strategy and strategic planning studies [5–7]. The mission statement is a component of strategic intent, but not synonymous with it; Ref. [8] defines strategic intent as a long-term perspective on

achieving a competitive stance, signaling change, a new direction, and a sense of purposeful destiny. Strategic intent aligns more closely with vision than with mission, which primarily focuses on defining an organization's enduring purpose, grounded in the present and extending indefinitely. Therefore, strategy guides the entire organization towards achieving the vision organization.

Despite mission statements being considered important by both academics and practitioners, there has been limited empirical research on mission statements in the past two decades [2,9,10]. Additionally, there is a shortage of research that has studied mission statements within the theoretical framework of sustainability [11]. Although considerable research has been conducted on the mission statements of significant global businesses, there is limited availability of studies that specifically examine the relationship between strategy and sustainability in mission statements. As sustainability becomes more important, mission statements become very important for connecting a company's purpose with long-term environmental, social, and governance (ESG) principles. This helps make sure that the strategies of companies are in line with changing global priorities.

There has been a big change in firm mission statements: shareholders used to focus on making as much money as possible in the short term, but now they focus on creating long-term value by promoting sustainability and incorporating environmental, social, and governance (ESG) principles. ESG is a way to measure how sustainable a company is [12]. It shows that better performance in environmental and social areas is linked to better financial results, like lower risk, lower costs of equity, and a better reputation [13]. Research, including meta-analyses of over 2000 studies, consistently demonstrates a positive relationship between ESG practices and financial performance, indicating that sustainability efforts benefit both companies and their stakeholders [13]. This shift highlights the growing recognition of ESG as a critical driver of long-term shareholder value, fostering better financial performance, stakeholder trust, and societal impact [14]. Also, environmental, social, and governance (ESG) principles are instrumental in advancing the United Nations Sustainable Development Goals (SDGs), providing a structured approach for companies to address global challenges such as poverty, inequality, climate change, and environmental degradation [15]. By prioritizing environmental stewardship, social equity, and sound governance, organizations contribute to clean energy, climate action, and societal well-being while fostering transparency, ethical decision-making, and long-term economic stability [16,17]. Central to this effort is a company's mission statement, which serves as a cornerstone of strategic planning by articulating its purpose, values, and long-term objectives. Mission statements not only align organizational efforts with sustainability goals but also embed ESG principles into the company's core identity, signaling commitment to stakeholders and enhancing resilience, reputation, and growth in an increasingly sustainability-driven global economy.

This study adds to the existing body of research on the relationship between the mission statement and ESG, which is significant in several different ways. First, developed countries like the United States and Europe have conducted the majority of research on mission statements and ESG [18]. Although Western models initially influenced mission statement initiatives in developing countries, it is important to note that the philosophy, culture, and societal values in Arab countries remain distinct and separate [19]. So, looking at mission statements and ESG in Palestine gives us useful information about a new country in the Arab region that is facing special problems, such as an unstable legal system, a war that is still going on, bad social and economic conditions, and institutional barriers. No previous studies have examined similar scenarios. This study also examines the impact of mission statement components and ESG in Palestine, a developing country characterized by high political and economic risks.

In order to obtain reliable and evidence-based results, it is crucial to conduct a comprehensive and well-designed study, considering the diverse, restricted, and often contradictory information on the influence of mission statements on sustainability. The aim of the current research was to proof the potential beneficial relationships between mission statements and firm sustainability, while also enhancing comprehension of strategy development within companies. The study will concentrate on these follow-up questions:

- Is there a relationship between mission statements and ESG in Palestinian companies?
- What are the typical components found in mission statements of Palestinian companies, and do these statements have ideal components?
- What components need to be integrated within mission statements? How significantly does the incorporation of specific components affect the sustainability of a business?
- Is it possible for managers of a company to obtain guidance on developing sustainable business strategies by examining the connection between mission statements and ESG?

The literature review findings are presented in Section 2. The research methodology and procedures are described in Section 3. The findings and results of the analyses are discussed in Section 4. The discussion results are in Section 5. The conclusion, implications, and limitations, as well as future research, are provided in Section 6.

## 2. Literature Review

### 2.1. Mission Statements: Purpose and Components

Companies officially recognize "mission statements" as official statements that define their core purpose, goals, and principles [20]. They offer advice and direction, assisting in aligning business efforts and resources towards common objectives. Additionally, mission statements play an important role in providing guidance for strategic decision-making as they aid management in recognizing and evaluating opportunities and challenges in connection with the organization's core values and goals [21].

Mission statements were the subject of extensive study in the 1990s [22]. However, the integration of commercial and social goals by new types of organizations, including benefit companies and social enterprises, has reignited interest in this field. Researchers such as [22,23] have made substantial contributions to the progress of both the theoretical and empirical aspects of mission statement research in recent years.

Mission statements are now considered a crucial tool in strategic management. They serve to articulate an organization's purpose, as stated by [24]. However, there are ongoing debates regarding the differences and effects of mission statements, visions, and objective declarations on organizations, as discussed by [25–27]. At times, mission statements contain additional declarations such as values, vision, purpose, business creeds, or philosophies. Previous studies have examined different aspects of mission statements, including their main components, size, context, industry, and unique features [28]. Additionally, refs. [24,29] discovered a positive correlation between more comprehensive mission statements and higher-performing firms. This suggests that mission statements should include several essential components, such as identifying target markets and customers, specifying geographic domains, and stating company philosophy.

The components of mission statements within companies of various levels, including sizable, diverse businesses, small businesses, and medium-sized businesses, have been a topic of academic study [30–32]. These studies have also covered various industries, including industrial firms [28], tourism and hospitality [33,34], and nonprofit organizations [35].

Mission statements play an important role in management research since they provide important insights into an organization's strategic intent and culture [36,37]. They enable the evaluation of the degree to which a company's actions and choices fit with its declared goals and principles, as well as whether they comply with broader societal norms and

expectations [38]. Furthermore, mission statements serve as a standard for evaluating the performance and efficiency of an organization over time by providing a distinct reference point for tracking advancement towards organizational objectives [20].

The study by [29] classifies the different components of a mission statement into eight distinct categories according to their content. The key components of a business plan include (1) customers and markets; (2) the main products and service; (3) the geographic area; (4) the core technologies; (5) the company's survival, growth, and profitability; (6) the company's philosophy; (7) the company's self-concept; and (8) the public image of the firm. These were then improved by [24], resulting in a total of nine identifiable components, as outlined in Table 1.

**Table 1.** Desired components of an ideal mission statement, adapted from [24].

| No. | Component | Theme |
| --- | --- | --- |
| 1 | Customers | Who are the business's potential customers? |
| 2 | Product/services | What are the company's primary offerings? |
| 3 | Markets | Where is the company a competitor? |
| 4 | Technology | What core technologies does the company use? |
| 5 | Survival, growth, and profitability | How driven is the company to achieving its economic goals? |
| 6 | Philosophy | What are the firm's primary sentences, core values, aims, and philosophical top priorities? |
| 7 | Self-concept | What are the primary competitive advantages and capabilities of the company? |
| 8 | Public image | What is the company's reputation? |
| 9 | Employees | What is the company's perspective on its employees? |

Additionally, [39] conducted further research by analyzing mission statements from 44 industrial companies and found 25 separate components. Only 11 of these components found a common use. These include the organization's mission or reason for being; its values, beliefs, or philosophy; a statement of its unique skills or strengths; an intended competitive position; a list of important stakeholders; overall corporate objectives or goals; one clear and convincing goal; specific target customers or markets; an emphasis on employee well-being; a focus on shareholder interests; and a vision statement.

In his study, [20] argues that our comprehension of the complete benefits of formulating and conveying a company's strategy through mission statements is still lacking. Although there are theoretical models, there is very limited actual proof to support their effectiveness. Prior research on mission statements has demonstrated that when they are effectively formulated and implemented, they have a positive impact on a company's performance, values, ethics, and stakeholders. The framework by [20] includes an analysis of the various components of the mission statement, the stakeholders mentioned, and the goals or intended purpose of the statement.

Recent discussions have examined the potential influence of mission statements on company performance. Bart's 1997 study examined the impact of involving different stakeholders in a company's goal statement on its financial performance. The results primarily show an unfavorable association, with the exception of a positive influence when employees were explicitly considered. This study expanded upon the findings of [29], who indicated that higher-performing organizations tend to possess components such as company philosophy, self-concept, and public image more frequently. However, research conducted by [31] in the specific context of Irish SMEs failed to confirm the assertion that these elements improved performance. In their study, ref. [39] found that there is a negative correlation between the presence of financial objectives in mission statements and the performance of a corporation. In a more recent study, ref. [20] examined the

mission statements of Fortune Global 500 companies. They found a noteworthy positive relationship between performance indicators (such as return on assets and return on sales) and the incorporation of employees and society in the mission statements. Nevertheless, no substantial correlation was discovered with other stakeholder groups, like employees, investors, and suppliers.

*2.2. Sustainability Mission Statement in the Previous Studies*

Contemporary scholarly discussions have begun to examine the concept of sustainability missions, with the first research studies concentrating on business missions, vision statements, and strategies [40]. These studies examine the manner in which organizations portray themselves to the public and manage their internal operations. Scholars have employed the phrase "sustainability missions" to refer to business missions that integrate sustainability objectives and associated endeavors [41]. The discussion surrounding sustainability missions also explores the construction of narratives within sustainability policy initiatives.

At first, the literature discussed sustainability missions in a general and unclear way. These discussions were found in studies of water management in China [42], environmental policy in Australia [16], and sustainability networks that involve multiple organizations [43]. These initial references did not indicate a cohesive comprehension or strategy. The concept gained traction after Mazzucato's influential research on mission-oriented innovation policies (MOIPs) [44].This research not only explored the practical implications but also prompted critical analysis of how firms can balance social, environmental, and economic values [45].

The literature has transitioned from emphasizing "technical missions" to "transformative missions", which employ a systemic strategy to bring about change, surpassing mere technological solutions. The broader perspective is reinforced by a focus on governance, multi-level coordination, and public participation [26,43].

Companies' corporate social responsibility (CSR) strategies involve sustainability and sustainable practices, which include economic, environmental, and social aspects [46,47]. Moreover, technological improvements have a crucial impact on sustainable progress [48], emphasizing the growing importance of technology as a major topic. Another crucial domain involves a rethinking of production and consumption patterns to support the sustainable economy [49].

*2.3. The Correlation Between Mission Statement and ESG*

Over the years, the mission statements of corporations have significantly evolved from a focus on short-term profit maximization [50], as emphasized by shareholder theory, to a broader approach of long-term value maximization for the wider society, aligning more with stakeholder theory [13]. This shift towards sustainable, long-term-oriented value creation is underscored by changes in investor behavior and market valuations. For instance, as of 2019, 84% of the S&P 500 companies' value was in intangible assets like reputation and customer loyalty, reflecting a departure from the traditional emphasis on physical assets [51]. This transformation also mirrors a broader societal expectation that corporations should contribute to the well-being of society and the environment, not just focus on financial gains [52].

Furthermore, the integration of (ESG) elements into business strategies and investment decisions highlights a diminishing trade-off between short-term returns and long-term value [53]. ESG metrics serve as concrete measures of a company's sustainability and social responsibility, influencing financial and operational performance positively. Academic studies, such as the meta-study by [13], support this positive correlation, showing that

firms with good sustainability implementations directly achieve better financial outcomes. This trend is reinforced in corporate mission statements that increasingly incorporate sustainability-related themes, suggesting a growing recognition of the significance of ESG elements in creating long-term value for all stakeholders, including shareholders [54,55].

Also, mission statements have evolved to serve as a key driver of organizational behavior, especially in the context of environmental, social, and governance (ESG) principles. Companies that focus their mission on ESG impacts are required to "walk-the-talk", as inconsistencies between stated commitments and actual actions can lead to significant employee backlash and reputational damage that is difficult to recover [12]. When firms prioritize short-term profit over their ESG commitments, it can erode employee trust and long-term engagement. Conversely, firms with shareholder value-focused missions have more flexibility, facing fewer penalties for prioritizing profits over ESG but gaining similar rewards when prioritizing the opposite [56]. However, adopting an ESG-focused mission narrows the decision set for companies, compelling them to consistently uphold their ESG commitments or risk long-term consequences.

Furthermore, the relationship between mission statements and ESG highlights the strategic framing of ESG investments [52]. Firms may emphasize altruistic motives, such as societal benefits, or align them with financial goals, including market demands and regulatory requirements. The way these investments are framed influences both internal stakeholders, such as employees, and external stakeholders, like customers and investors. Firms that position their ESG initiatives as societal contributions can enhance their reputation and mitigate potential negative outcomes if ESG efforts fail to produce long-term shareholder value [17]. Future research opportunities lie in exploring the long-term impacts of ESG-focused missions on employee behavior, the framing of ESG initiatives, and the psychological effects of mission-inconsistent actions on stakeholders. These studies could provide valuable insights into how firms balance their ESG goals with shareholder expectations and how this balance shapes organizational success.

*2.4. Mission Statements and ESG from Different Industrial Perspectives*

Many research studies have looked at how mission statements differ across countries, industries, and organizations, showing how well they match up with environmental, social, and governance (ESG) factors. For instance, a study by [57–60] analyzed mission statements from firms in Europe, Japan, and the United States, revealing differences rooted in cultural and strategic priorities. Japanese firms emphasized employee motivation and societal contributions, reflecting their cultural focus on social morals and societal well-being—an alignment with the social dimension of ESG. Conversely, American and European firms prioritized economic advantage, with European companies placing a stronger emphasis on defining their geographic influence [61]. These distinctions illustrate the varying degrees to which firms integrate ESG components, particularly in governance and social aspects, as influenced by regional and cultural contexts. Similarly, a study by [62] compared mission statements from leading American and Chinese firms. While both nations evaluated resources similarly, Chinese companies emphasized innovation and societal growth alongside corporate development, reflecting an integrated approach to ESG principles [59]. In contrast, American firms focused more on consumers, products, and services, demonstrating a stronger orientation toward economic and product-driven goals [32,63]. Based on these results, ESG priorities may be different in different industries and countries. For example, Chinese companies seem to connect innovation more closely with social development. Further, a study of analyzed mission statements from American and Indian companies highlighted differences in stakeholder engagement [64]. Indian businesses assigned considerable importance to customers, irrespective of whether they operated in the

manufacturing or service sectors, underscoring a socially inclusive approach. In contrast, American businesses focused more on assessing public image and corporate philosophy, reflecting a governance-oriented perspective on ESG. The aviation industry provides another context for examining ESG integration in mission statements. Research by [65] analyzed 50 international airline companies and found a predominant focus on self-image (85%), values (77%), and consumers (70%). This reflects a strong alignment with the governance and social components of ESG, as these elements highlight transparency, corporate identity, and customer focus. Additionally, a follow-up study by [66] on 80 airlines reaffirmed the evolving nature of mission statements, influenced by internal and external pressures. Another study by [65] found that products/services (79.7%), clients (69%), and markets (58.2%) were the most commonly referenced components, indicating a shift in focus that may align with market-driven sustainability goals. Finally, ref. [67] examined the mission statements of 47 public universities in Spain using Drucker's framework [68] for non-profit organizations. The findings highlight the unique governance and social priorities of non-profit institutions, as they emphasize public service and long-term societal impact over purely economic objectives. These studies demonstrate that mission statements reflect varying degrees of ESG integration across different industries and regions. Cultural influences, sectoral priorities, and market demands all contribute to these variations, offering valuable insights into how firms and organizations approach sustainability and governance through their strategic declarations.

## 3. Methodology

The current research uses an exploratory approach, utilizing content analysis. This study also includes the collecting of secondary data by analyzing mission statements of chosen companies through their official websites. The identification of the statements as mission statements was predicated on specific references to these terms on a website, given the existence of overlapping definitions in the literature [69,70].

### 3.1. Sampling

This study used purposive sampling technique whereby the sample was obtained from the list of 49 companies listed on the Palestine Exchange (PEX). The listed companies were selected to facilitate access to comprehensive corporate data, which was important for this research from [71].

Data referring to the descriptive characteristics of the company were obtained from their websites (legal form, social objective, founding date, location, sector, and number of employees). The five categories (financial, insurance, investment, services, and manufacturing) were created based on the industry type, using a combination of selected listed companies as shown in Table 2.

**Table 2.** The characteristics of the sample.

|  | Number of Companies | % |
| --- | --- | --- |
| Industry |  |  |
| Service sector | 9 | 18% |
| Insurance sector | 11 | 23% |
| Financial sector | 8 | 17% |
| Manufacturing sector | 10 | 20% |
| Investment sector | 11 | 22% |
| Employment |  |  |
| Less than 20 | 13 | 26% |
| Between 21 and 49 | 17 | 35% |
| Above 50 | 19 | 39% |

This classification depends on [71].

*3.2. Analytical Procedures*

A co-word semantic network analysis was used in this study to find patterns in the way Palestinian businesses talk about sustainability. Co-word analysis (CWA) figures out how often two or more words appear together to find patterns and connections. This creates a two-way network that links companies through shared ideas. This method revealed three key themes: sustainability, technology, and production and consumption. Sustainability encompasses corporate social responsibility (CSR) strategies addressing economic, environmental, and social dimensions, while technology plays a crucial role in driving sustainable progress. The theme of production and consumption highlights the need to redesign models to support sustainability. This approach effectively mapped firms' focus areas, reflecting their contributions to sustainability discourse as shown in Figure 1.

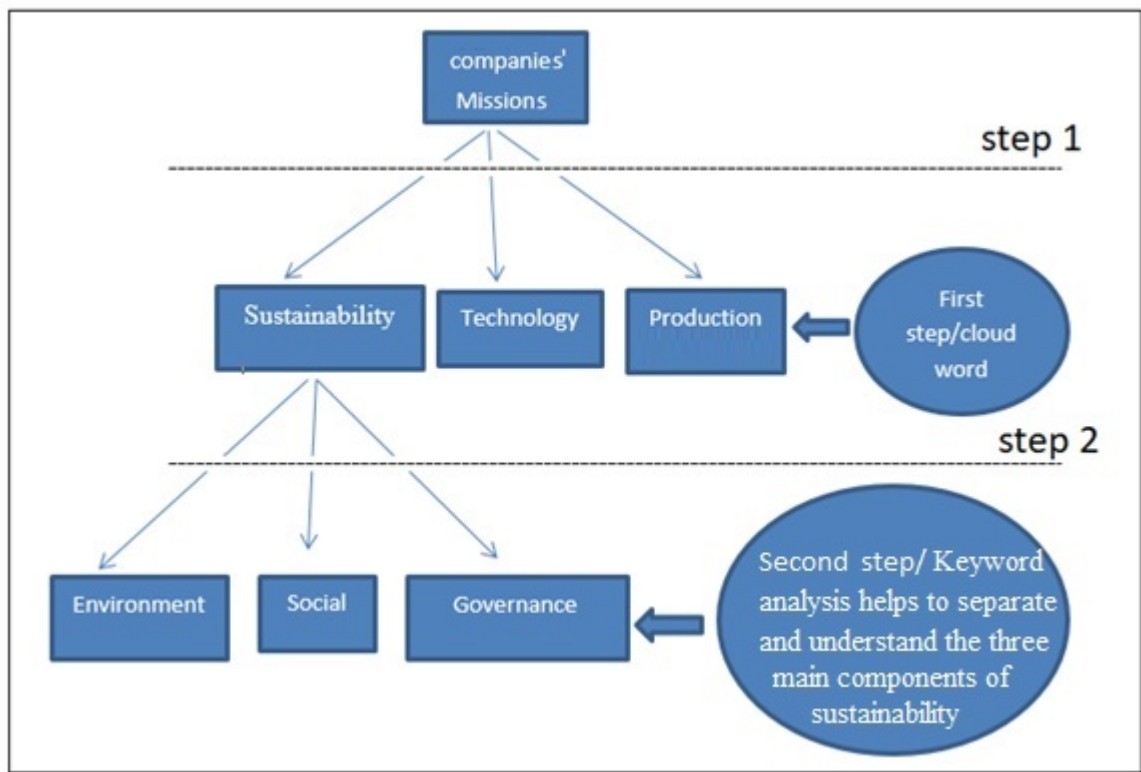

**Figure 1.** Framework of empirical study.

In the second step, the study looked at keywords to separate the three pillars of sustainability: social, economic, and environmental. It emphasized the need to include social aspects more in sustainability strategies. The analysis highlighted how Palestinian firms prioritize specific themes within sustainability and identified areas requiring more attention to accelerate the sustainable transition.

3.2.1. Content Analysis

Content analysis was utilized as a tool to measure mission statement components [69,70,72]; it is easier to form conclusions and spot patterns by organizing and systematically examining data. It facilitates researchers' comprehension of the framing and communication techniques. Researchers analyzed the content of each mission statement in depth, primarily qualitatively. This analysis aimed to determine the presence of the nine essential components that [24] suggested a good mission statement should have, as shown in Table 1; in this study, the stakeholder (like suppliers, society, and investors) component was added to the nine existing components, which was suggested by [73].

We used the MAXQDA software to evaluate the mission statements derived from qualitative data, making sure each label fit into the previously mentioned categories. More accurately, the program played a crucial role in avoiding data loss when conducting the examination of the data corpus. We allocated primary codes to either complete mission statements or parts of them. Subsequently, we organized these into second-order themes and further categorized them into sub-categories by consolidating the initial codes according to theoretical classifications. In conclusion, we combined these second-order themes according to the 10 categories shown in Table 3. The selected analytical framework specifies the main categories, also known as aggregate dimensions.

**Table 3.** The main categories form the mission statements.

| Categories | Sub-Categories |
|---|---|
| Customers (target market) | Consumers/clients/customers/businesses/small businesses/international businesses/people/youth/male/female/the government sector and NGOS. |
| Product/services | Features (manufactured, natural, etc.) and sorts (appliance, food, etc.) of goods/new/innovative products/services/design/collection/innovation/tailor-made services/consulting/luxury products and technological products/medical services. |
| Markets | Universal/international market/national market/local area/country/region. |
| Technology | Innovative/capacity/renewable/energy/speed/recovery resources/management solutions/waste/industrial service/project/efficiency/engineering/recycling/plants/reduce/collection/treatment. |
| Survival, growth, and profitability | Sustainability/growth and development/economical prices/the market share/advancing our company/investment/innovating/capital/corporation/competitiveness/production/new markets/limited liability/profit. |
| Philosophy | Cost leadership/quality/the best/social value/wellness/CSR/social/stakeholders' benefit/differentiation/well-being/common good/ environment/environmental protection/sustainable development/growth/ethics/quality production/innovation/leadership sector/social development/sustainability. |
| Self-concept | Sustainable technology/commitment to creativity/environmental stewardship/quality solutions. |
| Public image | Social/CSR/environment/technological solutions/innovation/reputation/transparency/accountability/leadership. |
| Employees | Support/team/recruitment/training/partners/staff/selection/career development/retaining. |
| Stakeholders | Suppliers/partners/investors/government/media/social/customers/employees/local institutions/environment. |

### 3.2.2. Word Cloud

The analysis of mission statements from all Palestinian companies in this study shows that sustainability, technology, and production are the main focal points. The word cloud generated for all Palestinian companies suggests that their mission statements include a broader range of targets, with a notable emphasis on environmental, social, and economic aspects, as shown in Table 3.

## 4. Findings

### 4.1. Analysis of Mission Statement Content "Cloud Words"

The main emphasis of the current study was on the "explanation" component of firm mission statements. Researchers used a classification approach to assign codes to the words in each sentence, resulting in a grand total of 22,323 words. We consolidated all the contents into a unified document and thoroughly revised it to eradicate any grammatical and punctuation errors, as well as to merge compound phrases by employing an underscore.

We designed the coding process using the software MAXQDA. These guidelines were used: (1) the text should consist of entire words that have significance, such as nouns and adjectives; (2) the analysis should eliminate words such as papers, words, prepositions, and adverbs; (3) the analysis should remove commonly utilized words, usually referred to as theme words; and (4) the analysis should give priority to terms that occur three times or more to enhance their visibility.

Using the semantic network they found, the researchers grouped words that appeared together frequently to find subjects. This method is similar to the one used by [61]. The team used the weighted modularity clustering strategy to account for the strength (i.e., link value) of connections between words, in accordance with the conventional method for text clustering recommended by [74]. Figure 2 shows the outcomes. We determined each word's weight based on the strength of its connections, also known as graph strength, which led to network significance.

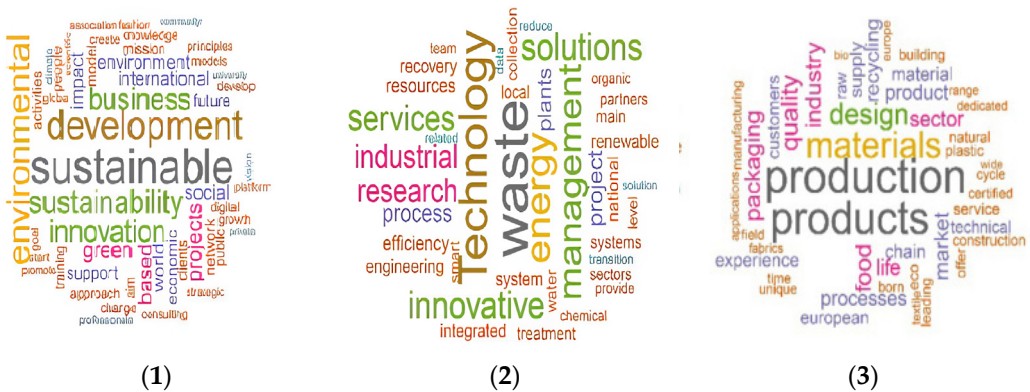

**(1)** **(2)** **(3)**

**Figure 2.** "Word cloud" analyzed for mission statement of the listed companies considering (**1**) sustainability, (**2**) technology, and (**3**) production.

As previously mentioned, three main topics emerged: (1) sustainability, (2) technology, and (3) production. The essential phrases "environment", "development", "natural", "projects", "innovation", and "economic" specifically express the three basics of sustainability (ESG). In addition, the terms "studies", "innovation", and "leadership" were found to be associated with technological themes. Production was primarily defined by the terms "manufacturing input" and "output", along with issues related to supply management.

Table 4 presents a sample of mission statements from various firms, illustrating their alignment with the three identified themes: sustainability, technology, and production. These mission statements show how well firms' stories fit into thematic groups and show how important these themes are to circular economy practices. For instance, sustainability, as seen in Company A's mission statement, involves the strategic pursuit of long-term economic and cultural progress for humanity and the environment. Similarly, technology plays a transformative role in the circular economy, as exemplified by Company B, which innovates biogas production from cow manure, addressing energy efficiency and resource recovery. Lastly, production processes, as highlighted by Company C, integrate sustainable manufacturing principles to ensure quality and circularity. These examples show how sustainable practices, new technologies, and production methods that use resources efficiently can all work together to help reach the goals of the circular economy. Many businesses are trying to follow the Sustainable Development Goals (SDGs), especially SDG 12 [75]. This is because sustainability is an important part of economic changes. Achieving these goals requires collaboration across supply chains and active engagement from both businesses and consumers [76].

**Table 4.** Example company mission statements for each category.

| Company | Mission Statement | Theme |
|---|---|---|
| A | "The goal is to speed the transition to good economic and cultural models by carefully strategizing our current actions for the long-term well-being of both humanity and the environment." | Sustainability |
| B | Palferm is a pioneering company that focuses on creating facilities for processing cow manure. These facilities transform manure into biogas, which can be used to create electricity. Additionally, Palferm explores the potential of recovering all the biological components from the manure. | Technology |
| C | "Our goods are crafted from high-grade repurposed denim, utilizing vegetarian and environmentally friendly components. They are carefully produced to ensure premium quality. The whole manufacturing process is characterized by transparency and is conducted using sustainable production methods and principles of the circular economy." | Production |

*4.2. The Degree of Environmental, Social, and Governance Aspects of Sustainability in Listed Companies*

The preceding section highlighted the categorization of sustainability into three pillars [56]. Therefore, we investigated the frequency of these basic components at the beginning of the debate discussion. By using computer software to further analyze the theme structure, we chose the MAXQDA program to determine the frequency of each pillar in the business descriptions: a keyword analysis.

An analysis was conducted to examine the frequency of each basic term in the company descriptions, using the software methodologies outlined by [77]. The most commonly used words were "community" and "human" in relation to the social aspect, "environment" and "nature" in relation to the environmental aspect, and "company" and "economic" in relation to the economic aspect. To ensure adaptability, we conducted the analysis again, this time using only a single keyword for each theme (namely, "social", "environment", "economic"). The outcomes were comparable; Figure 3 illustrates the frequency of business mentions related to each pillar. The analysis revealed that social references comprised the majority (46%) of the total, while economic references constituted a smaller proportion (35%) and references totaled around 22% for governance.

The second stage of the research proved to be quite valuable, as it allowed for the examination of sustainability, which is commonly separated into its three basic components. At first, the environmental aspect became the most important, emphasizing the need to protect environments, mankind, and nature from damaging activities.

Moreover, sustainability includes economic potential as well. Indeed, a sustainable long-term perspective necessitates the presence of economic prosperity, adequately paid employees, and equitable income distribution [36]. Therefore, although enterprises should not only prioritize profit, civil economy models show that implementing sustainability practices can lead to increased profitability; CSR theory argues for this approach [47,55]. In general, incentives are designed to make sustainable activities more competitive in comparison to traditional approaches. Applying taxation that is directly linked to the environmental impact of activities improves the overall effectiveness of such measures [41]. Failure to conform to fair competition principles may result in businesses being unable to operate effectively, potentially leading to a global phenomenon. In addition to these economic factors, social factors are equally significant. The ethical aspect is sometimes undervalued due to its lack of a direct correlation with any measurable effect [38].

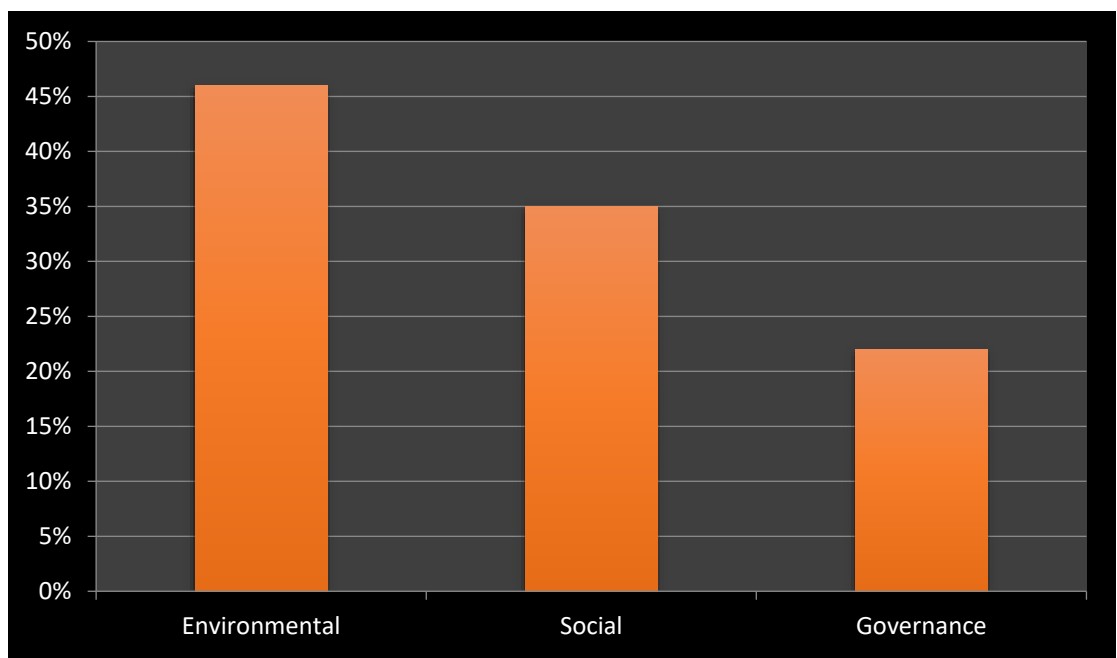

**Figure 3.** Frequencies of the social, economics, and governance aspects across listed firms.

*4.3. Integration of ESG Components in Corporate Mission Statements: Insights from Key Priorities*

Figure 4 provides an illustration of a percentage of companies that have integrated various components into their mission statements, in line with the Environmental, Social, and Governance (ESG) criteria. Many companies prefer their customers, with a majority of 90% placing a strong emphasis on them. Additionally, 85% of companies focus on their products and services, indicating a dedication to ensuring customer satisfaction and maintaining high product quality. These values match with the social dimension of ESG, which emphasizes consumer protection and product responsibility [16,53]. Approximately 82% of companies prioritize markets, indicating a strong emphasis on strategic market positioning and competitive behavior. This focus is crucial for ensuring transparency in governance and promoting fair competition [46]. With 76% of companies accepting technology, it is clear that they are committed to environmental innovations that focus on reducing pollution and improving resource efficiency, which are crucial for long-term sustainability [40]. In a similar vein, it is worth noting that 82% of companies prioritize survival, growth, and profitability. This demonstrates their commitment to financial sustainability and effective management practices, which are crucial for maintaining good governance [9,13,48]. Philosophy and self-concept, commonly discussed by many companies, play a crucial role in shaping culture, ethics, and compliance within an organization [16]. Nevertheless, the relatively lower importance placed on public image (64%), employees (49%), and stakeholders (57%) implies that these aspects may not be given as much priority in mission statements. This could indicate potential shortcomings in addressing comprehensive ESG strategies, particularly in terms of engaging internal stakeholders and fostering public trust. Overall, it is worth noting the strong focus on customer orientation and market competitiveness. However, there is an opportunity for improvement in terms of considering environmental factors and engaging with a wider range of stakeholders. This would help correspond mission statements with broader ESG objectives.

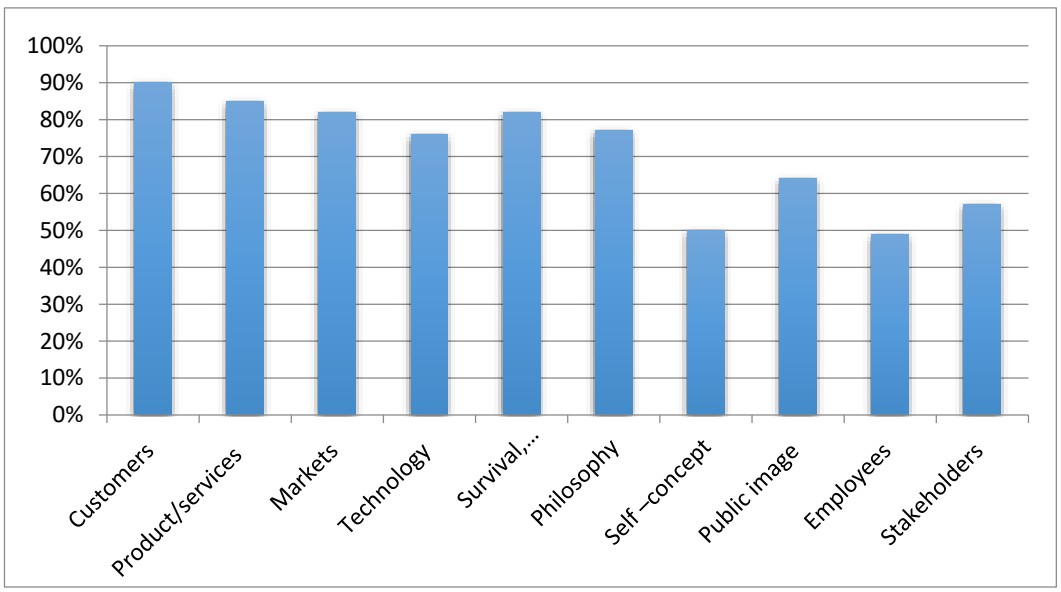

**Figure 4.** Availability of mission components in the mission statement.

*4.4. Comparative Representative ESG Among Categories Sectors*

Figure 5 presents an illustration of the manner in which various industry sectors prioritize certain components of their mission statements. This provides insights into the degree to which these sectors correspond with environmental, social, and governance (ESG) criteria. A high focus is placed on "Customers" and "Product/Services" across all industries, which indicates a common dedication to satisfying the requirements of consumers and providing excellent goods. This is a crucial component of the social portion of ESG considerations. On the other hand, there is a significant disparity in the emphasis placed on technology. The manufacturing sector places a significantly greater emphasis on this component than the investment and finance sectors do. This suggests that manufacturing may be more in line with environmental sustainability through technological innovation.

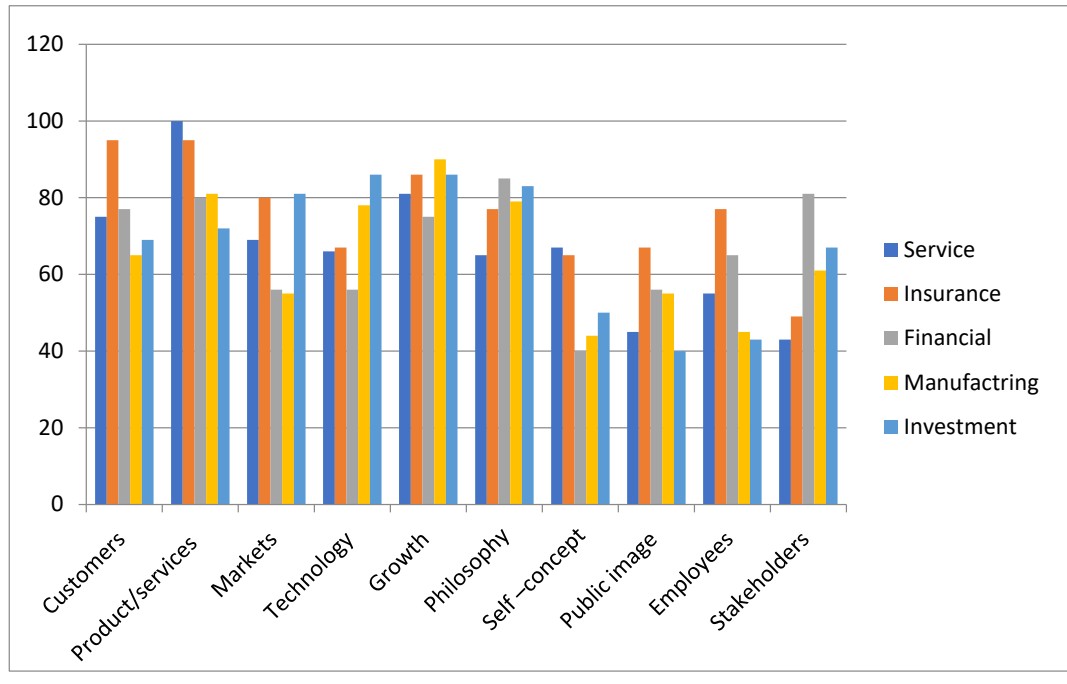

**Figure 5.** Representative mission components among sectors.

A further point to consider is that industries such as investment place a significant emphasis on stakeholders, which is closely aligned with the ESG's emphasis on ethical governance and larger social impact. This shows that industries that place a priority on stakeholders may be better aligned with ESG principles, particularly in the manner in which they manage relationships and the well-being of their employees, which contributes to the sustained viability of the organization over the long term [38,40,78,79]. Overall, the data from the figure demonstrate a complicated interaction between the components of the mission statement and the principles of environmental, social, and governance aspects (ESG). Each industry demonstrates distinct priorities that represent the specific challenges and opportunities they face when attempting to incorporate ESG into their strategic goals.

## 5. Discussion

The academic literature on strategic management highlights the critical role of mission statements in shaping corporate governance, defining organizational goals, and communicating values to stakeholders. This study agrees with [80,81] that there is a significant gap in the way sustainability is talked about in mission statements. The primary objective of this research was to evaluate the extent to which Palestinian companies embed sustainability issues into their mission statements, focusing specifically on how these companies express their commitment to ESG.

An analysis of 49 mission statements from Palestinian companies reveals that sustainability, technology, and production are prominent themes, as demonstrated by word cloud analysis. This indicates that these companies possess a significant awareness of sustainability issues, with these themes being integral to their strategic objectives despite the political and economic challenges associated with the Palestinian situation. Supporting this conclusion highlights the importance of production and technology in fostering sustainability and enabling environmental adaptation [82]. This study shows that environmental problems and societal needs are very closely linked by looking at how mission statements and environmental, social, and governance (ESG) criteria fit together; this answers the first research question.

The findings further demonstrate the substantial focus Palestinian companies place on ESG issues, particularly in addressing environmental challenges, given that the Palestinian economy is subject to Israeli policies that control the land, especially the control of natural resources such as water and natural resources. This is consistent with [76]'s research, which advocates integrating environmental considerations into corporate philosophies to enhance sustainability. However, the study also identifies notable gaps in the mission statements analyzed. The most frequently emphasized component, present in 90% of the mission statements, was "customers", reflecting a strong focus on addressing societal needs and expectations [14]. Additionally, 82% of mission statements prioritized survival, growth, and profitability, indicating a balanced approach to economic, social, and environmental objectives. These findings effectively address the second research question by identifying common components within the mission statements.

Contrarily, the least frequently mentioned "employees" component suggests a lack of emphasis on the human aspect of business sustainability due to the lack of adequate laws that protect Palestinian employees, in addition to the lack of sufficient opportunities to work in the Palestinian market, as the Palestinian market is relatively small and closed. According to [12], this omission could have adverse implications for long-term organizational performance. To achieve a more comprehensive ESG alignment, mission statements should more effectively address employees, stakeholders, and public image. Incorporating these components would enhance employee satisfaction, foster stakeholder trust, and improve public perception, thereby contributing to long-term sustainability. This

insight addresses the third research question by highlighting areas for improvement in mission statement composition.

Examining the connection between mission statements and ESG provides valuable insights for managers. Managers can make strong plans that balance economic, social, and environmental concerns by finding gaps and making sure that mission statements are in line with overall ESG goals. This alignment not only supports sustainable business practices but also provides actionable guidance for managers, thereby addressing the fourth research question.

Moreover, the emphasis on customer satisfaction as a revenue driver is evident across various industries in the Palestinian business landscape. The manufacturing sector, for instance, demonstrates a strong focus on expansion, technological innovation, and environmental progress, which is attributed to the presence of sufficient Palestinian talent in the Palestinian market and their ability to understand and understand quality issues. This proactive engagement underscores the sector's commitment to business sustainability and its efforts to address broader ESG challenges. The focus on specific industries further brings out how different industries have different strategic priorities. These priorities are shaped by the unique problems and goals that companies in each sector face.

## 6. Conclusions

### 6.1. Conclusion

The study employed content analysis to examine the mission statements of 49 Palestinian businesses, focusing on key themes such as production, technology, and sustainability. We utilized advanced qualitative analysis tools, such as MAXQDA software and word cloud techniques, to comprehensively analyze the textual data. The results show that environmental, social, and governance (ESG) principles are a big part of the mission statements. This shows that the companies are committed to sustainability and doing business in a responsible way. However, significant gaps remain in the holistic integration of ESG initiatives, particularly regarding internal stakeholder engagement and building public trust. To better align corporate strategies with ESG objectives, the study recommends prioritizing environmental innovation and fostering greater stakeholder participation.

To enhance long-term sustainability, we encourage Palestinian companies to strike a balanced integration of internal and external components in their mission statements. Regular evaluation of the effectiveness of strategic planning processes is essential to ensuring clear organizational direction and the improved achievement of business goals. Incorporating ESG as a competitive advantage is critical, with particular emphasis on the active involvement of stakeholders and management, especially in developing countries. To fill in the gaps that have been found, more research should be conducted using larger samples, longitudinal studies to see how things change over time, and quantitative methods to learn more about how ESG factors work in different industries, how well environmentally friendly practices work, and corporate social responsibility frameworks.

### 6.2. Implication

Based on the findings described above, this study provides key implications for the mission statement and ESG literature through managerial and academic insights.

Managerial Implications: Our findings suggest that ESG initiatives among the selected companies are still at a basic level, lacking comprehensive due diligence and strategic development. The majority of the study's companies still struggle with customer acquisition, survival, growth, and profitability, indicating a need to expand the scope of their mission statement components. Managers should address the misconception that a mission statement is a slogan raised for public consumption. In addition, this study can help com-

panies gain a broader understanding of MS systematically, which is essential for enhancing ESG, encouraging businesses to develop clear ESG, institutionalizing best practices, and expanding ESG disclosure requirements.

Academic Implications: The findings contribute to the ESG literature by providing empirical support for uncertainty environments in the context of Palestine. The study confirms that internal and external environments influence ESG practices. This relationship highlights the importance of clarifying the concept of MS in motivating companies toward ESG activities that align with the interests of shareholders and stakeholders.

### 6.3. Limitations and Future Research

This study is subject to several limitations that must be acknowledged. First, a key limitation arises from the small sample size, as the research focuses on Palestinian firms, a context constrained by the country's small size and unstable political environment. The limited number of listed companies in Palestine further restricted the scope of the study. Second, there is a scarcity of prior research on this topic in the Arab world, and particularly in Palestine, which poses challenges in building a strong comparative or theoretical foundation for the study. Future research could address these limitations by analyzing a larger and more diverse sample as the number of Palestinian companies increases. Using research methods that help us understand mission statements better, like interviews or qualitative analyses, could lead to more useful information. We could also add mission statements to other data sources like indicators or sustainability reports which provide tangible evidence of how these aspirations translate into measurable actions and outcomes. By aligning the discourse in mission statements with the data and metrics presented in sustainability reports, organizations can demonstrate accountability, build stakeholder trust, and ensure that their practices align with their stated values.

**Author Contributions:** Data curation, A.A.; investigation, A.A.; methodology, A.A. and K.D.; supervision, K.D.; writing—original draft, A.A. and I.Y.Y.A.; writing—review and editing, A.A. All authors have read and agreed to the published version of the manuscript.

**Funding:** This research received no external funding.

**Institutional Review Board Statement:** Not applicable.

**Informed Consent Statement:** Not applicable.

**Data Availability Statement:** The original contributions presented in the study are included in the article; further inquiries can be directed to the corresponding author.

**Conflicts of Interest:** The authors declare no conflicts of interest.

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
