# Peer review of "How Strategic Planning Enhances ESG: Evidence from Mission Statements"

_sustainability, doi:10.3390/su17020595_

Round 1

Reviewer 1 Report

Comments and Suggestions for Authors

Abstract

The objective statement needs to be rewritten to clearly represent the purpose of this paper. In general, the abstract should briefly present an introduction to the topic, the purpose of the work, the method used, and the results clearly. It is recommended that the authors review these points. I also suggest that the authors submit the paper for a language review.

1. Introduction

There is no theoretical foundation for the information presented in the first paragraph. The gap identified by the authors needs to be better justified in the introduction section. The authors should elaborate more on the limitations they found in the literature. Citation 9 could be removed without harming the content of the paper. Additionally, it would be helpful if the authors explained more clearly why they chose Palestinian companies.

2. Literature Review

Section 2.3 could be more thoroughly described, as it is the central point of this paper.

3. Materials and Methods

There are grammatical errors in Figure 1 that need to be corrected. The errors throughout this section hinder the clarity and interpretation of the paper.

4. Findings

The analysis presented in Table 4 shows ambiguity. The authors need to clearly demonstrate how they determined that, in the second case, the chosen category was "technology" rather than "environmental" or "circular economy." Subsection 4.3, titled "availability ESG," does not have a clear objective. The title of this subsection needs to be rewritten.

5. Discussions

The discussions are a central point of this paper. The suggestion is for the authors to rewrite this section to directly address the questions raised at the beginning of the paper. The same applies to the conclusion section, where the authors should clearly present how the conclusions of the paper contribute to advancements in both theory and practice. It is also necessary for the authors to indicate the limitations of the study, such as whether the results apply only to Palestinian companies, the significance and relevance of these companies in the region, and what concrete actions should be taken based on the results of this study.

I hope these suggestions contribute to improving the paper. Congratulations on the effort!

Comments on the Quality of English Language

The paper needs to undergo a language review throughout all its sections. Sections, subsections, tables, and figures need to be revised.

Author Response

Dear Editor,
Thank you for taking the time to review and edit my manuscript, "Strategic Planning Enhances ESG: Evidence from the Mission Statement," submitted to the Sustainability journal. We greatly appreciate your constructive feedback and insightful comments, which have been invaluable in enhancing the quality and clarity of our work. Your expertise and thorough review have contributed significantly to improving the manuscript, and we are grateful for your efforts and time Below, we have also provided a point-by-point response explaining how we have addressed each comment. We are looking forward to the outcome of your assessment.

Yours sincerely.

Response to Reviewer 1 Comments

1.    Point-by-point response to Comments and Suggestions for Authors

Comments 1: (Abstract)

The objective statement needs to be rewritten to clearly represent the purpose of this paper. In general, the abstract should briefly present an introduction to the topic, the purpose of the work, the method used, and the results clearly. It is recommended that the authors review these points.

We thank the reviewer for his constructive comment. We have improved the writing of the abstract.

Comments . 2 ( Introduction)
There is no theoretical foundation for the information presented in the first paragraph. The gap identified by the authors needs to be better justified in the introduction section. The authors should elaborate more on the limitations they found in the literature. Citation 9 could be removed without harming the content of the paper. Additionally, it would be helpful if the authors explained more clearly why they chose Palestinian companies.

Thanks for the comments. We have added references to enhancing the theoretical foundation in the first paragraph. We justified the gap in the introduction in paragraph four, and we also elaborate on the limitations of the literature for this study. We have also removed citation 9 from the introduction. We are also justifying why we chose Palestinian companies.

Comments . 3 (Literature Review)

Section 2.3 could be more thoroughly described, as it is the central point of this paper.

Thanks for the comments. We have cited many recent papers in the revised manuscript to improve the literature review and make the research gap much clearer.

Comments .4  (Materials and Methods)

There are grammatical errors in Figure 1 that need to be corrected. The errors throughout this section hinder the clarity and interpretation of the paper.

Thanks for the comments. We correct the grammatical error in figure 1, and we also rewrite the paragraph to be clear.

Comments .4  (Findings)

The analysis presented in Table 4 shows ambiguity. The authors need to clearly demonstrate how they determined that, in the second case, the chosen category was "technology" rather than "environmental" or "circular economy." Subsection 4.3, titled "availability ESG," does not have a clear objective. The title of this subsection needs to be rewritten.

Thank you for bringing this point to our attention. We added an explanation regarding Table 4 and justified that. We also rewrote the title of subsection 4.3.

Comments .5  (Discussions)

The discussions are a central point of this paper. The suggestion is for the authors to rewrite this section to directly address the questions raised at the beginning of the paper. The same applies to the conclusion section, where the authors should clearly present how the conclusions of the paper contribute to advancements in both theory and practice. It is also necessary for the authors to indicate the limitations of the study, such as whether the results apply only to Palestinian companies, the significance and relevance of these companies in the region, and what concrete actions should be taken based on the results of this study. 

We thank the reviewer for his constructive comment. We rewrote this section to address the questions of the study. Also, we arrange the last section regarding the conclusion, and we develop this section for implications and limitations.

Response to Comments on the Quality of English Language

We are sorry for the inconvenience caused. We have addressed all the mentioned points. We hope the new version is free from these mistakes. In addition, we have proofread the manuscript with the assistance of an English teacher.

Reviewer 2 Report

Comments and Suggestions for Authors

Please see the detailed comments in the attached document.

Author Response

Response to Reviewer 2 Comments

Dear Editor,

Thank you for taking the time to review and edit my manuscript, "Strategic Planning Enhances ESG: Evidence from the Mission Statement," submitted to the Sustainability journal. We greatly appreciate your constructive feedback and insightful comments, which have been invaluable in enhancing the quality and clarity of our work. Your expertise and thorough review have contributed significantly to improving the manuscript, and we are grateful for your efforts and time Below, we have also provided a point-by-point response explaining how we have addressed each comment. We are looking forward to the outcome of your assessment.

Yours sincerely.

Point-by-point response to Comments and Suggestions for Authors

Comments (1) 
The paper refers to limited research on mission statements within the framework of sustainability theory. It is recommended that the authors compare and analyze the findings of existing literature on the relationship between mission statement and ESG to identify the innovations and differences in this study. Analyze the differences in ESG elements in company mission statements across industries or regions and explore the reasons behind these differences. 

We thank the reviewer for his constructive comments. We have added section 2.4 to the literature review, which explains how mission statements and ESG are implemented across various industries worldwide, along with the reasons for these differences.

Comments (2)
Palestinian listed companies were selected for the paper. It is recommended to add a comparative analysis with firms from other countries or regions to enhance the generalizability and persuasiveness of the findings.

We thank the reviewer for the insightful comment. We're adding a comparison between countries in Section 2.4 to the literature review and how differences between cultures affect them.

Comments (3)
It is proposed to elaborate on the gap in existing research on the relationship between mission statements and ESG and how this study fills this gap.

We thank the reviewer for his comment. We added this suggestion in paragraph five in the introduction and added in paragraph six to explain this gap. 

Comments (4)

It is suggested that the authors could explicitly state the relevance of the study, such as the importance of ESG in the current global SDGs and the central role of mission statements in a company's strategic planning.

We thank the reviewer for his comment. We added this suggestion in third and fourth paragraph in section 2.3 of the literature review 

4. Response to Comments on the Quality of English Language

We are sorry for the inconvenience caused. We have addressed all the mentioned points. We hope the new version is free from these mistakes. In addition, we have proofread the manuscript with the assistance of an English teacher.

Round 2

Reviewer 1 Report

Comments and Suggestions for Authors

Abstract

The abstract is more concise and well-written. However, the authors need to ensure that the results align with the objective: "The research aims to identify the key components of these mission statements and assess how effectively they integrate sustainability concepts." As currently written, the research results do not reflect the stated objective.

Introduction

There are some formatting and font size errors throughout the introduction that need to be corrected. The first sentence of the second paragraph needs to be rewritten for greater conciseness and objectivity. The second paragraph could be rewritten as a whole, focusing on conciseness and clarity. Additionally, a smoother transition is needed between the topics of sustainability and mission statements.

From citation 17 onward, there are no more citations in this section of the introduction. The information presented here should be well-supported by references to ensure the relevance of the work.

A very positive point is the justification for the choice of Palestinian companies. The angle the authors explored regarding uncertainty scenarios, coupled with Arab culture, is highly relevant and could be further developed.

Theoretical Review

The authors have significantly improved the theoretical framework section and clearly included the relationship between ESG and mission statements. The only suggestion is for the authors to reduce the length of the paragraphs to improve readability.

Material and Methods

Figure 1 still needs correction. The first block labeled "Mission’s Companies" should be "Companies’ Missions."

Where it says "cloud word," the correct term is "Word Cloud."

The resolution and quality could also be improved. The font size should be standardized across the figure. The same applies to the title of section 3.2.2.

Discussions

Please find a substitute term for "big hole" used in the first paragraph of this section.

In this section, the authors could better explore the angle of uncertainty conditions that they introduced in the introduction.

Conclusions

In this section, the authors could simply title it as “Conclusions.”

Again, the authors could further explore the angle presented in the introduction.

I do not believe that the use of software combined with the interpretation of a researcher constitutes a limitation of this work.

I also understand that the promotional nature of mission statements is not a limitation, but rather a factor to consider.

A very relevant path that should be highlighted is the one presented at the end of the paragraph: adding information and comparing the discourse in the mission statement with reality through reports and indicators.

The authors should review the entire paper for formatting and spacing errors.

Congratulations to the authors for their effort and the corrections they have made. I hope these suggestions further improve the work. My position is that after these minor revisions, the paper should be ready for publication, especially due to the angle introduced in the introduction.

Author Response

Dear Editor,

Thank you for taking the time to review and edit my manuscript, "Strategic Planning Enhances ESG: Evidence from the Mission Statement," submitted to the Sustainability journal. We greatly appreciate your constructive feedback and insightful comments, which have been invaluable in enhancing the quality and clarity of our work. Your expertise and thorough review have contributed significantly to improving the manuscript, and we are grateful for your efforts and time Below, we have also provided a point-by-point response explaining how we have addressed each comment. We are looking forward to the outcome of your assessment.

Yours sincerely.

Comments 1: (Abstract)

The abstract is more concise and well-written. However, the authors need to ensure that the results align with the objective: "The research aims to identify the key components of these mission statements and assess how effectively they integrate sustainability concepts." As currently written, the research results do not reflect the stated objective.

We thank the reviewer for his constructive comment. We have improved the writing of the abstract to be consistent with the body of the paper.

Comments . 2 ( Introduction)

There are some formatting and font size errors throughout the introduction that need to be corrected. The first sentence of the second paragraph needs to be rewritten for greater conciseness and objectivity. The second paragraph could be rewritten as a whole, focusing on conciseness and clarity. Additionally, a smoother transition is needed between the topics of sustainability and mission statements.

From citation 17 onward, there are no more citations in this section of the introduction. The information presented here should be well-supported by references to ensure the relevance of the work.

A very positive point is the justification for the choice of Palestinian companies. The angle the authors explored regarding uncertainty scenarios, coupled with Arab culture, is highly relevant and could be further developed.

Thanks for the comments regarding the introduction. The formatting and the font size were corrected. We are rewriting the second paragraph to make it more accurate. Additionally, we are incorporating a single sentence into the fourth paragraph to establish a connection between sustainability and our mission statement.

Additionally, we have included a citation in paragraph six to bolster our work. Many thanks for appreciating our efforts.

Comments . 3 (Literature Review)

The authors have significantly improved the theoretical framework section and clearly included the relationship between ESG and mission statements. The only suggestion is for the authors to reduce the length of the paragraphs to improve readability.

Thanks for the comments. We removed some paragraphs without contradicting the theoretical framework.

Comments.4 (Materials and Methods)

Figure 1 still needs correction. The first block labeled "Mission’s Companies" should be "Companies’ Missions." Where it says "cloud word," the correct term is "Word Cloud."

The resolution and quality could also be improved. The font size should be standardized across the figure. The same applies to the title of section 3.2.2.

Thanks for the comments. We correct the grammatical error in figure 1, and we also corrected the term "Word Cloud" in this section. Also, we corrected the font size for the whole paper.

Comments .4  (Discussions)

Please find a substitute term for "big hole" used in the first paragraph of this section.

In this section, the authors could better explore the angle of uncertainty conditions that they introduced in the introduction.

We thank the reviewer for his constructive comment. We paraphrase the term "big hole" into "significant gap." And also added more explanation to paragraphs to be consistent with the introduction of the paper.

Comment .5 (conclusion)

In this section, the authors could simply title it as “Conclusions.”

Again, the authors could further explore the angle presented in the introduction.

I do not believe that the use of software combined with the interpretation of a researcher constitutes a limitation of this work.

I also understand that the promotional nature of mission statements is not a limitation but rather a factor to consider.

A very relevant path that should be highlighted is the one presented at the end of the paragraph: adding information and comparing the discourse in the mission statement with reality through reports and indicators.

The authors should review the entire paper for formatting and spacing errors.

Thanks for the comments regarding the conclusion. We changed the title to "conclusion" and remain the subtitles as they are. We removed the third and fourth limitations as suggested. We have also included information about sustainability reports in the document. We have corrected the formatting and font size throughout the entire paper.
